# Dynamic Retrieval Augmented Generation Based on The Knowledge-Aware of Large Language Models

## Abstract

Retrieval-augmented generation (RAG) has proven effective in enhancing open-domain question answering by supplementing language models with external knowledge. However, current approaches often rely heavily on the model's internal confidence scores to decide whether retrieval is necessary. This overreliance, coupled with the tendency of language models to show overconfidence, results in excessive and sometimes redundant retrieval operations. Moreover, conventional RAG workflows commonly incorporate coarse-grained retrieved documents directly into the generation process, which introduces noise and semantic drift that can compromise answer quality. To address these limitations, we propose DynaRAG, a dynamic knowledge-aware framework built on a three-stage optimization strategy. First, a hybrid question-knowledge similarity space is constructed, along with a lightweight threshold prediction network that learns query-adaptive decision boundaries to control retrieval triggering more precisely. Second, we generate multigranularity semantic variants to perform targeted retrieval and rank documents using a newly introduced knowledge importance scoring mechanism, thus improving the relevance and specificity of the retrieved content. Third, a prompt-guided large language model synthesizes the final answer based on the refined input of selected knowledge. Extensive experiments demonstrate that DynaRAG achieves average improvements of approximately 11% in EM and 14% in F1 on six representative QA benchmarks. Evaluated against a diverse suite of retrieval-augmented baselines, DynaRAG consistently improves both accuracy and efficiency, underscoring its robustness and adaptability in knowledge-intensive tasks.

## 1 Introduction

Large language models (LLMs) have recently achieved impressive results in natural language understanding, generation, and reasoning tasks (Brown et al., 2020). Despite this progress, recent studies have shown that state-of-the-art models still suffer from hallucination-generating content that is factually incorrect or semantically inconsistent (Zhang et al., 2024).This issue stems from the limitations in timeliness and completeness of the knowledge stored within the model parameters (Zhou et al., 2023).

Retrieval-Augmented Generation (RAG) has emerged as a powerful paradigm for enhancing large language models (LLMs) with access to external, non-parametric knowledge sources (Zhu et al., 2024). Unlike purely parametric models (Brown et al., 2020) that rely solely on static internal representations, RAG frameworks retrieve relevant documents or passages from external corpora and condition the generation process on this additional evidence. This design offers several advantages: it mitigates hallucinations by grounding responses in verifiable knowledge, enables models to adapt to domain-specific or rapidly evolving information, and improves performance in knowledge-intensive tasks such as open-domain question answering, fact verification, and multi-hop reasoning (Islam et al., 2024; Izacard & Grave, 2021). Recent developments have further extended RAG to specialized domains such as biomedical information retrieval, legal document analysis, and multimodal reasoning, demonstrating its broad applicability and flexibility (Chen et al., 2024; Fang et al.,

2024; Xu et al., 2024a). Despite these advances, current RAG systems continue to face two fundamental challenges: (1) determining when external retrieval is necessary (Lewis et al., 2020) and (2) effectively integrating heterogeneous, often noisy knowledge sources (Asai et al., 2024). Addressing these issues is critical for improving both the reliability and scalability of RAG-based intelligent applications.

In terms of retrieval decision-making, most current approaches rely on the model's internal confidence to decide whether external retrieval is necessary (Izacard & Grave, 2021). However, this confidence-typically derived from token-level likelihoods-reflects statistical co-occurrence rather than a metacognitive assessment of knowledge reliability (Yu et al., 2024). As a result, models may trigger unnecessary retrievals or not retrieve essential information when facing novel inputs or knowledge gaps. Moreover, when retrieved content includes contradictory evidence, standard linear integration strategies can exacerbate error propagation. About knowledge integration, conventional RAG pipelines typically employ a cascading architecture that lacks robustness against retrieval noise (Izacard et al., 2023). These systems often treat full documents as indivisible units, overlooking the varying information density within them (Zhang et al., 2024). Empirical studies show that only about 25% of the sentences in a retrieved document contribute meaningful information (Huang et al., 2025). This coarse-grained treatment results in two issues: it increases the likelihood of incorporating irrelevant content and dilutes the impact of critical knowledge fragments (Su et al., 2024).

In this paper, we propose a dual-module optimization framework for RAG. First, a dynamic retrieval mechanism is introduced, which leverages activation pattern analysis to construct a fine-grained problem–document relevance matrix. By employing prompt-based knowledge representation and adaptive thresholding, this mechanism dynamically regulates retrieval boundaries, enabling context-sensitive and selective retrieval. Second, we design a multi-dimensional knowledge distillation mechanism that enhances document utilization through semantic query expansion and document confidence re-ranking. Specifically, this module:(a) generates semantically diverse query variants, (b) performs parallel retrieval to collect candidate documents, (c) applies re-ranking algorithms to calculate document importance, and (d) integrates high-confidence evidence into the generation process. Together, these modules enhance both the precision of retrieval triggering and the robustness of knowledge integration. Our study aims to provide a comprehensive solution to the challenges in retrieval-augmented generation, improving the overall effectiveness and reliability of knowledge integration in LLMs. The performance of DynaRAG was assessed using benchmark datasets spanning single-hop, multi-hop, and knowledge-intensive QA tasks. The empirical findings indicate that our framework delivers significant improvements in accuracy while reducing redundant retrieval, thereby achieving a better balance between effectiveness and efficiency in real-world applications.

## 2 RELATED WORK

Building on recent advances in LLMs, two areas of research have become particularly relevant to this study. The first concerns retrieval-augmented generation (Lewis et al., 2020), which integrates external knowledge retrieval into generative models to enhance factual accuracy and contextual relevance. The second focuses on instruction-following (Lou et al., 2024), which aim to improve a model's ability to understand and execute natural language instructions in a reliable and controlled manner. The following subsections review prior work in these two domains.

### 2.1 RETRIEVAL-AUGMENTED GENERATION

Large language models (LLMs) often hallucinate when handling domain-specific or knowledge-intensive queries. RAG (Lewis et al., 2020) addresses this limitation by grounding responses in external knowledge and has become a widely adopted paradigm in QA and conversational systems (Ren et al., 2025). Existing methods can be grouped into several categories: **query-based** approaches append retrieved passages to the input (Shi et al., 2024; Lewis et al., 2020; Asai et al., 2024), **representation-based** methods integrate evidence at the latent level (Izacard & Grave, 2021; Borgeaud et al., 2022), **logic-based** approaches couple retrieval with decoding via probabilistic alignment (Xiong et al., 2021; He et al., 2021), and **inference-based** methods replace or complement generation with retrieval (Lan et al., 2023; He et al., 2024). Recent extensions further explore

active retrieval during decoding (Jiang et al., 2023; Wang et al., 2023a), leveraging internal model memory (Liu et al., 2024; Sun et al., 2023), task-specific retrievers (Jeong et al., 2024; Yu et al., 2023), and retrieval optimization (Xu et al., 2024b; Sarthi et al., 2024). Despite these advances, challenges remain in mitigating noisy or conflicting evidence, maintaining coherence in multi-turn interactions, and ensuring scalability for real-time applications.

### 2.2 INSTRUCTION FOLLOWING

Instruction following (Lou et al., 2024) enables LLMs to interpret natural language instructions and generate task-consistent outputs across QA, summarization, and reasoning. Early progress came from prompt-based reformulations and in-context learning (Brown et al., 2020), followed by lightweight adaptation methods such as prompt and prefix tuning (Lester et al., 2021; Li & Liang, 2021). A major breakthrough was instruction tuning with curated datasets (e.g., FLAN (Wei et al., 2021), InstructGPT (Ouyang et al., 2022)), often combined with reinforcement learning from human feedback (RLHF) to align outputs with user expectations. Subsequent work scaled synthetic instruction data (Honovich et al., 2023; Köksal et al., 2024; Wang et al., 2023b) and introduced alternative alignment strategies such as preference optimization (Wang et al., 2024). While these methods improve generalization and robustness, they remain limited in knowledge-intensive settings. Recent studies have explored combining retrieval with instruction-following models (Lin et al., 2024; Zhu et al., 2024), highlighting the need for tighter integration between instruction interpretation and dynamic retrieval to enhance factual grounding and adaptability.

## 3 METHOD

As illustrated in Figure 1, we present the design of our framework, denoted as DynaRAG, which comprises three components: Knowledge Reliability Evaluator (KRE), Knowledge Distillation Enhancer (KDE), and Retrieval-Augmented Generation. KRE dynamically assesses the necessity of document retrieval through probabilistic thresholding. The KDE module refines the retrieved documents through knowledge distillation processes. RAG integrates the relevant documents with the input query through the LLMs, later generating the final answer.

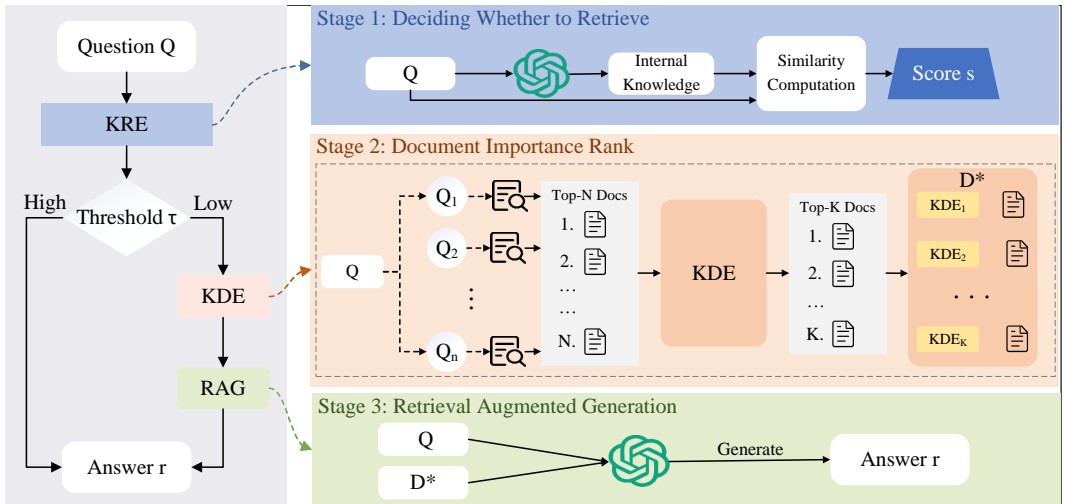

Figure 1: Overview of DynaRAG

### 3.1 KNOWLEDGE RELIABILITY EVALUATOR

The Knowledge Reliability Estimator (KRE) implements fine-grained retrieval timing control through a three-stage cascaded knowledge evaluation mechanism. This method is used through the following computational pipeline: First, it extracts the internal knowledge representation vectors of the LLM via prompt engineering techniques, thereby constructing a cross-modal alignment

space between the query and candidate documents through hybrid similarity measurement. Then, a lightweight threshold prediction network generates query-adaptive dynamic decision boundaries by learning the latent distribution characteristics of knowledge relevance. Finally, directed document retrieval is triggered based on the comparative analysis between the computed knowledge similarity score and the predicted adaptive threshold.

**Internal Knowledge Extraction** We design a structured prompt template "Answer strictly based on existing knowledge: {Q}" and feed it into the LLM, thereby extracting its internal knowledge representations through forward propagation analysis.

**Similarity Metric Computation** We construct a hybrid similarity metric that simultaneously quantifies semantic space alignment (via the cosine term) and generation distribution divergence (via the KL divergence term). The hybrid similarity metrics are reformulated, as shown in Eq.1.

$$s(q, d_j) = \frac{\cos(h_q^{\text{int}}, h_{d_j}^{ext})}{T_{temp}} + \lambda \cdot KL\left(p_{LM}(q)//p_{LM}(d_j)\right) \tag{1}$$

where the cosine part captures surface-level semantic matching between query and document embeddings, while the Kullback-Leibler Divergence part(Eq.2) finds latent knowledge conflicts by contrasting the LLM's generation distributions conditioned on the query versus the knowledge document. $p_{LM}(\cdot|x)$ is the probability distribution of the vocabulary of the model under the input $x$.

$$KL\left(p_{LM}(q)//p_{LM}(d_j)\right) = \sum_{w \in V} p_{LM}(w+q) \log \frac{p_{LM}(W+q)}{p_{LM}(W+d_j)} \tag{2}$$

**Adaptive Threshold Prediction** To enable personalized retrieval personalization, we develop a context-sensitive threshold prediction framework using pre-trained language model embeddings. This method replaces fixed retrieval thresholds with dynamically computed boundary conditions derived through semantic analysis of input queries, effectively mitigating domain-specific bias inherent in conventional static threshold approaches. The proposed mechanism is used by projecting query semantics into a latent decision space where retrieval necessity is determined via adaptive boundary estimation.

**Model Framework** The input query is first processed using the BERT tokenizer, which performs subword segmentation. The tokenized sequence is then fed into a pretrained BERT model for contextual encoding. The encoder produces a sequence of hidden states across multiple layers, capturing hierarchical linguistic representations. From the final hidden layer, the hidden state corresponding to the [CLS] token is extracted. This vector serves as a condensed semantic representation of the entire query, effectively summarizing its global meaning through self-attention. To transform the extracted features, a multi-layer perceptron (MLP) with two hidden layers is employed. The final output layer, activated by a sigmoid function, yields a continuous value within the (0,1) interval, representing the predicted threshold.

**Training** A small-scale training and validation dataset is constructed, with each sample annotated by the degree to which the corresponding query depends on external knowledge. The sampling and annotation criteria are detailed in Table 7. Sentence representations are obtained using a pre-trained language model (BERT), from which fixed-length embeddings are extracted. These embeddings are passed through a two-layer MLP, followed by a sigmoid activation to normalize the output within the range (0,1). The model is optimized using MSE loss. To mitigate overfitting, the BERT encoder is kept frozen during training, and only the MLP head is updated.

## 3.2 Knowledge Distillation Enhancer

To mitigate potential retrieval bias caused by single-query formulations, we propose a multi-perspective retrieval augmentation strategy. Specifically, a set of semantically similar queries is constructed based on the original input. For each query in this set, independent retrieval is performed to obtain a corresponding set of relevant documents. These documents are then ranked using a relevance scoring method we proposed. Finally, the original query, along with the aggregated and ranked list of relevant documents, is fed into an LLM to generate the final answer.

**Rag Fusion** Given an original query, we design a structured prompt template to guide the generation of diverse yet semantically equivalent reformulations. The prompt is as follows: "Generate {n} questions that are semantically equivalent to, but phrased differently from, the

following question: q." For each generated similar query, a separate retrieval process is performed to obtain a corresponding list of relevant documents. Concretely, similar question generation and relevant document retrieval are expressed in Eq.3 and Eq.4.

$$q_i' = LLM(T_{prompt}(q); \tau, k) \tag{3}$$

$$D_i = Retriever(q_i'; top - n) = \{d_{ij}\}_{j=1}^n \tag{4}$$

**Rank** The significance of retrieved documents depends not only on the generation confidence but also on token importance, token semantics, and the influence of each token on subsequent tokens. To address the limitations of existing methods, we propose a novel approach for evaluating the importance of retrieved documents, named Knowledge Distillation Enhancer (KDE). This method assesses the relevance score by quantifying not only the uncertainty associated with each token,but also its semantic contribution and contextual influence on subsequent content. KDE begins by quantifying the uncertainty of each token within the retrieved document. This is achieved by computing the entropy of the token's probability distribution over the vocabulary. For a given retrieved document, its content is treated as a sequence denoted by $T = \{t_1, t_2, \ldots, t_n\}$, where each $[t_i$ represents the token at position i in the sequence. For any token $t_i$ , the entropy $H_i$ is computed as Eq.5.

$$H_i = \sum_{v \in V} p_i(v) \log p_i(v) \tag{5}$$

Here, $p_i(v)$ denotes the probability of generating token v from the vocabulary at position i. This entropy-based measure of uncertainty constitutes the first dimension in our multi-faceted evaluation of the token significance.

**KDE** further leverages the inherent self-attention mechanism in Transformers to assign weights to tokens, reflecting their influence on subsequent context. Specifically, for any given token $t_i$ , its contextual impact is quantified by recording the maximum attention weight $a_{max}(i)$. This is derived from the final layer of the LLM's Transformer, capturing the highest attention score associated with $t_i$ . The influence of a token $t_i$ is determined by identifying the highest attention score $a_{max}(i)$ among all positions $j > i$ , $a_{max}(i)$ is computed as Eq.6. The attention score between tokens $t_i$ and $t_j$ , denoted as $A_{i,j}$ , is computed as Eq.7.

$$a_{max}(i) = \max_{j>i} A_{i,j} \tag{6}$$

$$A_{i,j} = softmax(\frac{Q_i K_j^T}{\sqrt{d_k}}) \tag{7}$$

Here, $Q_i$ denotes the query vector of token $t_i$ , and $K_j$ represents the key vector of token $t_j$ . The attention score $A_{i,j}$ is computed by applying the softmax to the scaled dot product of $Q_i$ and $K_j$ , normalized by the square root of the key dimension $d_k$. To account for the semantic contribution of each $t_i$ , KDE employs a binary semantic criterion to filter out stopwords, thereby focusing on tokens $t_i$ with significant semantic value. The detailed information is computed as shown in Eq.8.

$$s_i = \left\{ \begin{array}{l} 0, \text{if} t_i \subseteq S \\ 1, otherwise \end{array} \right. \tag{8}$$

Here, S denotes the set of stopwords,and $s_i$ represents the semantic contribution score of $t_i$ . By integrating uncertainty, contextual importance, and semantic value, KDE computes a composite score for each $t_i$ . The detailed representation is obtained by the calculation in Eq.9.

$$S_{KDE}(t_i) = H_i \cdot a_{max}(i) \cdot s_i \tag{9}$$

Here, $T_j$ denotes the set of tokens in document $d_j$ . Documents are then ranked based on the aggregate scores $S_{KDE}(d_i)$ of their tokens, resulting in an ordered list of retrieved documents. The detailed information is computed as shown in Eq.10.

$$S_{KDE}(d_i) = \frac{1}{|T_j|} \sum_{t_i \in d_j} S_{KDE}(t_i) \tag{10}$$

**Inference** Given the filtered document set $D_{final}$ and the original query $q$ , the prompt template is designed as follows:"Please answer the following question based on the relevant documents and their associated relevance scores:d1(score1), d2(score2), d3(score3), ... Question: q"
Finally, the prompt will be input into the large model to obtain the final answer. Specifically, the detailed information is calculated according to Eq.11.

$$A_i = LLM(T_{prompt}(q) =; \tau, k) \tag{11}$$

## 4 EXPERIMENTS

### 4.1 EXPERIMENTAL SETUP

#### 4.1.1 BASELINES

We compare our method with representative RAG baselines across three categories. **Parametric-only** models (e.g., Direct) rely solely on the LLM's internal knowledge. **Static RAG** methods, such as Standard RAG, REPLUG (Shi et al., 2024), and SURE (Kim et al., 2024), follow fixed retrieve-then-generate pipelines without runtime adaptation. **Dynamic RAG** methods, including IRCoT (Trivedi et al.), FLARE (Jiang et al., 2023), Self-RAG (Asai et al., 2024), and SKR (Wang et al., 2023a), interleave retrieval with generation for stepwise reasoning. Finally, **Adaptive RAG** approaches, such as Adaptive-RAG (Jeong et al., 2024), RQ-RAG (Chan et al., 2024), and Iter-RetGen (Shao et al.), adjust retrieval strategy or query formulation based on task complexity or intermediate outputs. This taxonomy ensures coverage of major retrieval paradigms. Detailed information is provided in the Appendix B.

#### 4.1.2 EVALUATION DATASETS

We evaluate on widely used open-domain QA benchmarks to cover single-hop, multi-hop, and long-tail knowledge scenarios. Specifically, we adopt **Natural Questions (NQ)** (Kwiatkowski et al., 2019), **TriviaQA** (Joshi et al., 2017), and **WebQuestions** (Berant et al., 2013) for general open-domain QA; **HotpotQA** (Yang et al., 2018) and **2WikiMultiHopQA** for multi-hop reasoning; and **PopQA** (Mallen et al., 2023) for long-tail factual coverage. Performance is measured using Exact Match (EM) and F1, following standard QA evaluation protocols.Detailed information is as shown in the Appendix A.

#### 4.1.3 EXPERIMENTAL SETTINGS

**Stopword** We employ the en_core_web_sm model from the SpaCy NLP toolkit for stopword detection within the KDE module. This lightweight yet efficient model is widely recognized for its robust performance in practical NLP pipelines, and its reliability has been validated in prior work (Shelar et al., 2020).

**Knowledge Bases for RAG** For all datasets, Wikipedia was used as the retrieval knowledge base. The collection consists of passages extracted from English Wikipedia, a widely adopted source in benchmark datasets. The corpus construction involved downloading Wikipedia snapshots in XML format, removing HTML tags, extracting clean textual content, and segmenting the text into retrieval-ready passages.

**Adaptive Threshold Prediction Settings** To estimate the external knowledge dependency of each query, we construct a small-scale training/validation dataset labeled with knowledge reliance scores. Sentence embeddings are extracted using a frozen pre-trained BERT encoder to prevent overfitting, followed by a two-layer MLP head with a sigmoid activation for score normalization. The model is trained using mean squared error (MSE) loss. Sampling and labeling criteria are defined as shown in Table 2 and Table 3.

### 4.2 RESULTS

As shown in Table 1, DynaRAG consistently demonstrates superior performance across all six benchmark datasets, outperforming both traditional and state-of-the-art retrieval-augmented generation methods. Compared to recent adaptive and iterative approaches such as RQRAG and Iter-RetGen, DynaRAG exhibits stronger reasoning ability and retrieval efficiency, particularly on benchmarks like NQ and HotpotQA. On the challenging 2WikiMultiHopQA dataset, which requires multi-hop and cross-document reasoning, DynaRAG sets a new performance standard, validating the effectiveness of its dynamic retrieval mechanism in complex reasoning scenarios. In comparison to standard RAG, DynaRAG demonstrates enhanced integration of structured knowledge and improved answer accuracy on both short-form factual queries and longer open-domain questions. Against adaptive baselines like Adaptive-RAG, DynaRAG shows greater robustness to semantic ambiguity

and more reliable performance in fuzzy or uncertain contexts. Furthermore, on knowledge-intensive datasets such as WebQA, DynaRAG overcomes prior limitations in long-tail entity recognition, reflecting its stronger generalization across domains and knowledge distributions.

Table 1: Performance comparison of DynaRAG and baseline methods across six benchmark datasets.

| Method | NQ EM / F1 | TriviaQA EM / F1 | WebQA EM / F1 | Hotpotqa EM / F1 | 2WikiQA EM / F1 | PopQA EM / F1 |
|---|---|---|---|---|---|---|
| Direct | 28.7 / 36.1 | 56.9 / 62.4 | 16.4 / 21.8 | 17.8 / 21.6 | 18.2 / 22.3 | 28.5 / 33.8 |
| Standard RAG | 33.7 / 41.3 | 57.1 / 64.2 | 15.3 / 21.4 | 30.3 / 34.9 | 16.9 / 20.8 | 30.5 / 35.1 |
| REPLUG | 26.7 / 41.5 | 55.9 / 62.7 | 18.6 / 25.0 | 25.0 / 29.7 | 15.3 / 18.6 | 21.7 / 26.0 |
| Sure | 34.6 / 49.4 | 51.8 / 58.3 | 22.8 / 29.1 | 26.2 / 31.3 | 15.4 / 18.8 | 41.2 / 47.0 |
| IRCoT | 31.5 / 34.6 | 54.1 / 60.5 | 19.7 / 26.3 | 35.1 / 40.0 | 25.3 / 29.6 | 37.8 / 43.3 |
| FLARE | 21.3 / 30.9 | 53.2 / 59.1 | 17.723.8 | 21.6 / 25.9 | 26.3 / 31.1 | 15.2 / 18.9 |
| Self-RAG | 33.6 / 47.0 | 35.9 / 42.8 | 19.2 / 25.7 | 22.8 / 27.6 | 19.4 / 23.5 | 25.7 / 30.5 |
| SKRknn | 30.6 / 38.0 | 53.1 / 59.4 | 14.4 / 20.2 | 26.3 / 31.2 | 17.1 / 20.5 | 25.3 / 30.2 |
| Adaptive-RAG | 33.9 / 41.6 | 55.4 / 62.3 | 14.3 / 20.5 | 32.1 / 37.2 | 21.6 / 25.5 | 32.1 / 37.6 |
| RQ-RAG | 31.0 / 39.4 | 50.1 / 56.7 | 23.7 / 30.2 | 27.8 / 32.0 | 28.3 / 32.9 | 39.1 / 45.0 |
| Iter-RetGen | 34.5 / 47.9 | 58.4 / 65.3 | 16.1 / 21.9 | 32.4 / 37.3 | 15.4 / 18.9 | 30.7 / 35.5 |
| **DynaRAG** | **36.1 / 50.6** | **61.2 / 66.7** | **25.7 / 38.9** | **36.5 / 41.2** | **28.9 / 33.7** | **43.1 / 48.7** |

## 4.3 ANALYSIS

Table 2: Comprehensive Evaluation of DynaRAG Model Components: Ablation Experiments, Retrieval Evaluation, Similarity Computation, and Comparison of Question Generation Strategies

(a) Ablation results on NQ and TriviaQA datasets, reporting EM scores.

| Method | NQ(EM) | TriviaQA (EM) |
|---|---|---|
| Direct | 28.7 | 66.9 |
| Standard RAG | 33.7 | 57.1 |
| No KRE | 34.0 | 57.5 |
| No KDE | 34.5 | 57.4 |
| **DynaRAG** | **36.1** | **59.5** |

(b) Performance comparison of different similarity computation methods

| Method | NQ(EM) | TriviaQA (EM) |
|---|---|---|
| BM25 | 28.9 | 48.3 |
| Sentence-BERT | 34.5 | 55.8 |
| Cross-Encoder | 35.0 | 56.7 |
| DPR | 33.8 | 54.2 |
| Contriever | 33.2 | 53.7 |
| Dragon | 30.5 | 50.1 |
| **DynaRAG** | **36.1** | **59.5** |

(c) Accuracy of different retrieval evaluation methods on the PopQA dataset.

| Retrieval Evaluator | Accuracy |
|---|---|
| **KRE** | **81.4** |
| ChatGPT | 56.7 |
| ChatGPT-CoT | 60.7 |
| ChatGPT-few-shot | 62.6 |

(d) Comparison of different question generation strategies within the KDE module

| Method | NQ(EM) | TriviaQA (EM) |
|---|---|---|
| Rule-based | 31.7 | 50.9 |
| Back-translation | 32.4 | 52.3 |
| SimCSE | 34.5 | 56.1 |
| **DynaRAG** | **36.1** | **59.5** |

### 4.3.1 MODULE ABLATION

To validate the effectiveness of the Knowledge Retrieval Enhancement (KRE) and Knowledge Distillation Enhancement (KDE) modules, we conduct comprehensive ablation studies across three benchmark tasks. As shown in Table 2a, DynaRAG consistently outperforms the base Standard RAG framework, demonstrating the complementary benefits of KRE and KDE in enhancing retrieval quality and answer accuracy. On HotpotQA, removing KRE significantly degrades performance, indicating its critical role in guiding multi-hop reasoning and path planning. The absence

of KDE leads to performance drops on TriviaQA, highlighting its effectiveness in filtering common sense noise and improving answer precision. Detailed task-wise analysis further supports these observations. On NQ, DynaRAG surpasses Standard RAG, confirming that both KRE and KDE jointly contribute to stronger structured knowledge integration. Removing either module results in clear performance degradation, whereas their combined presence yields synergistic gains-suggesting a positive coupling effect between retrieval quality and knowledge refinement. In HotpotQA, DynaRAG demonstrates substantial improvements in logical consistency and reasoning coherence. The dual-module setup effectively reduces fragmented or logically inconsistent predictions, showing its strength in modeling complex relational dependencies. While DynaRAG also outperforms Standard RAG on TriviaQA, it still trails behind the Direct method. This is primarily because Direct benefits from brute-force retrieval strategies, which are naturally suited to fact-based, well-defined queries such as numerical lookups. In contrast, DynaRAG's dynamic decision-making introduces slight overhead for simpler tasks. Future work will address this limitation by introducing lightweight module-switching mechanisms to adaptively handle such cases.

### 4.3.2 RETRIEVAL QUALITY EVALUATION

In Table 2c, we focus on evaluating the performance of retrieval quality assessment, a critical component in retrieval-augmented systems. Our lightweight DynaRAGKRE evaluator significantly outperforms general-purpose dialogue models such as ChatGPT and its enhanced variants using chain-of-thought prompting or few-shot learning. The results highlight that a retrieval-specific evaluation framework can more accurately distinguish relevant documents, demonstrating its crucial role in building efficient and reliable question-answering systems. This comparison underscores the necessity of specialized evaluators over generic models for precise retrieval quality judgment.

### 4.3.3 SIMILARITY METHOD ANALYSIS

To investigate the impact of various similarity computation methods on model performance, we conducted experiments using BM25, Sentence-BERT, Cross-Encoder, DPR, Contriever, and Dragon (Lin et al., 2023). As shown in Table 2b, DynaRAG retrieval framework consistently achieves superior results across open-domain QA (NQ), complex entity matching (TriviaQA), and multi-hop reasoning (HotpotQA) tasks. Its key advantage stems from innovative improvements in the retrieval mechanism. Compared to traditional methods, DynaRAG demonstrates clear gains in semantic understanding, precise multi-feature fusion, and dynamic decision-making for multi-hop inference. While deep models outperform conventional retrieval approaches, DynaRAG effectively balances retrieval granularity and computational efficiency, yielding stable improvements across tasks with controlled overhead.

### 4.3.4 QUESTION GENERATION METHODS ABLATION

In the KDE module, different question generation methods exhibit notable variations in retrieval performance. We conducted experiments comparing rule-based rewriting, back-translation augmentation, and semantic variant generation using a contrastive learning model (SimCSE). As shown in Table 2d, rule-based rewriting, constrained by a predefined synonym dictionary, showed the weakest results across open-domain QA (NQ), entity matching (TriviaQA), and multi-hop reasoning (HotpotQA), particularly underperforming on reasoning completeness in multi-hop tasks. The Back-translation improved entity matching accuracy by introducing cross-lingual semantic perturbations but yielded limited gains in complex reasoning scenarios. SimCSE, leveraging deep semantic encoding, achieved balanced improvements, demonstrating contrastive learning's effectiveness in capturing implicit semantic relations, though its static representations showed adaptation limits in dynamic contexts. Our proposed DynaRAG approach, utilizing large model prompting with structured templates, consistently outperformed these methods by generating high-quality semantic variants synergized with retrieval enhancement strategies.

### 4.3.5 HYPERPARAMETER ANALYSIS

To figure out the optimal number of generated similar questions (N), we conducted experiments on HotpotQA, 2WikiQA, and PopQA with varying N values. Figure 2 shows that all datasets achieve peak performance at N=3. Increasing N to this level enhances model robustness through greater se-

mantic diversity. However, beyond this point, performance systematically declines, suggesting that excessive generation introduces noise that disrupts the retrieval of key evidence chains. This trend highlights a trade-off between semantic augmentation and noise interference, with N=3 standing for a balanced choice for maximizing retrieval effectiveness and model stability.

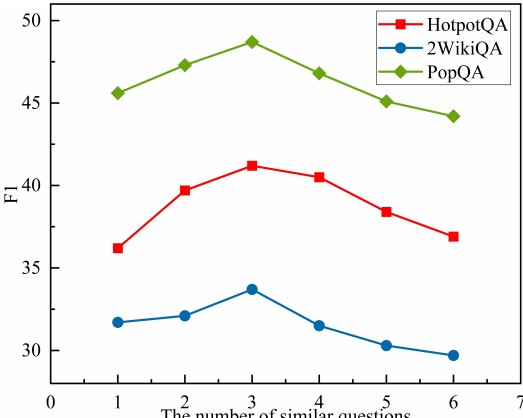

Figure 2: Impact of the number of generated similar questions (N) on retrieval performance across HotpotQA, 2WikiQA, and PopQA.

## 5 CONCLUSIONS

We presented DynaRAG, a dynamic knowledge-aware retrieval-augmented generation framework with a two-stage optimization design. The first stage employs a lightweight threshold prediction network to adaptively regulate retrieval by integrating large language model embeddings with a joint question–document similarity space. The second stage enhances retrieval quality through multi-granularity query reformulation and cross-document re-ranking, followed by contextual reasoning for final answer generation. Across five benchmark datasets spanning open-domain, multi-hop reasoning, and specialized tasks, DynaRAG consistently outperforms strong RAG baselines, indicating the value of adaptive retrieval for improving response accuracy and robustness.

Nonetheless, several limitations should be acknowledged. The threshold prediction module relies on careful dataset construction, which may constrain its ability to capture fine-grained, domain-specific nuances in highly specialized settings. In addition, the multi-perspective query expansion introduces extra computational cost, which could limit its practicality in latency-sensitive scenarios.

## 6 LLM USAGE

During the preparation of this work, the authors used OpenAI ChatGPT for language polishing and grammar correction. After using this tool, the authors reviewed and edited the content as needed and take full responsibility for all aspects of the publication.

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

## A  DETAILED INFORMATION OF THE DATASET

To comprehensively assess open-domain question answering (QA) systems across diverse reasoning requirements and data modalities, we adopted a set of widely recognized benchmark datasets. Natural Questions (NQ) and TriviaQA contain authentic user queries paired with Wikipedia-linked passages, capturing linguistic diversity and realistic retrieval conditions. WebQuestions focuses on question answering over structured knowledge bases, while HotpotQA and 2WikiMultiHopQA are designed to evaluate multi-hop and cross-document reasoning. PopQA examines coverage of long-tail factual knowledge. These datasets were selected to ensure balanced evaluation across single-hop, multi-hop, and knowledge-intensive QA scenarios.

**Natural Questions** (NQ) (Kwiatkowski et al., 2019)is a question-answering dataset consisting of real, anonymized queries issued to the Google search engine. Annotators are presented with a question and a Wikipedia page from the top 5 search results and label a long answer (typically a paragraph) and a short answer (one or more entities) if present. The dataset includes 307,373 training examples with single annotations, 7,830 development examples with five-way annotations, and 7,842 five-way annotated test examples.

**TriviaQA**(Joshi et al., 2017) is a large-scale reading comprehension dataset containing over 650K question-answer-evidence triples. It comprises 95K question-answer pairs authored by trivia enthusiasts, with supporting evidence documents collected independently. The dataset features complex compositional questions with high lexical and syntactic variability, requiring significant cross-sentence reasoning to identify correct answers.

**PopQA** (Mallen et al., 2023)is a large-scale open-domain QA dataset with 14K entity-centric question-answer pairs. Each question is automatically generated from a Wikidata knowledge triple using a template-based approach. Annotations include the subject entity, object entity, relation type, and Wikipedia page view statistics, providing a structured context for evaluating factual knowledge.

**WebQuestions** (WebQA) (Berant et al., 2013)is a QA dataset built using Freebase as the underlying knowledge base. It contains 6,642 question-answer pairs, where questions were gathered via the Google Suggest API and answers were obtained through crowdsourcing. In the original split, 3,778 examples are used for training and 2,032 for testing. All answers are mapped to Freebase entities.

**HotpotQA** (Yang et al., 2018)is a multi-hop QA dataset collected from English Wikipedia, comprising approximately 113K crowdsourced questions. Each question requires reasoning over two Wikipedia articles and is paired with two gold paragraphs and a list of supporting sentences identified by annotators as necessary for answering the question. **2WikiMultiHopQA** (2WikiQA) is a large-scale multi-hop QA dataset designed to assess reasoning and inference across multiple passages. Created by researchers at the National Institute of Informatics, it includes 192,606 examples. Questions are constructed to require multi-step reasoning, supported by both structured and unstructured evidence, with annotated reasoning paths to facilitate interpretability and evaluation.

**Metric** Each dataset is evaluated using both EM (Exact Match) and F1 metrics. EM measures strict string-level agreement between the predicted and reference answers, making it particularly informative for datasets with normalized, discrete answer spaces such as NQ, TriviaQA, and WebQuestions. F1, on the other hand, computes the harmonic meaning of precision and recall based on token overlap, providing a more flexible evaluation of partial matches. This is especially relevant for complex, multi-hop datasets such as HotpotQA, 2WikiMultiHopQA, and PopQA, where answers are more diverse and context-dependent.

## B DETAILED INFORMATION ON THE BASELINE

We compare our approach against a diverse set of baseline methods categorized by their retrieval strategies and integration with generation.

**Parametric-only models** (e.g., Direct) rely solely on the language model's internal knowledge without external retrieval.

**Static RAG**, such as Standard RAG, REPLUG, and SURE, incorporates retrieved evidence before generation in a fixed format without runtime adaptation.

Standard RAG adopts a fixed retrieve-then-generate paradigm. It uses a pretrained dense retriever to retrieve the top-k relevant documents from a corpus. These documents are concatenated with the original query and passed as input to a frozen LLM. The model generates answers based on this augmented context. This simple pipeline lacks adaptive control over retrieval or document integration.

REPLUG(Shi et al., 2024)treats the retriever as a modular component in the RAG pipeline. Each retrieved document is concatenated with the query independently and passed through the LLM. The final answer is obtained by weighted aggregation of all outputs based on retrieval similarity scores. REPLUG also introduces LM-supervised retrieval, aligning the retriever with the LLM by minimiz-

ing KL divergence between retrieval and model output distributions. This improves compatibility and performance without modifying the LLM.

SURE (Kim et al., 2024) introduces a candidate-conditioned summarization framework. The LLM first generates multiple candidate answers. For each candidate, a summary is generated from the retrieved documents to support that specific answer. The LLM then evaluates these summaries based on informativeness and supportiveness. The candidate with the strongest evidence is selected as the final answer.

**Dynamic RAG**, IRCoT, FLARE, Self-RAG, and SKR interleave retrieval with generation, allowing context-aware and stepwise evidence acquisition.

IRCoT (Trivedi et al.) interleaves retrieval and reasoning in a step-by-step manner. The model first generates an intermediate reasoning step, then retrieves supporting documents based on that step. This process is repeated, forming a closed loop of reasoning and retrieval. It enables multi-hop reasoning with targeted evidence. However, the method requires multiple LLM calls and incurs higher computational costs.

FLARE (Jiang et al., 2023) proposes a proactive retrieval mechanism during generation. The LLM incrementally generates sentences and monitors token-level confidence. When low-confidence tokens are detected, a retrieval is triggered and the output is revised using retrieved content. This dynamic approach integrates information as needed during decoding. It is particularly effective for long-form, knowledge-intensive generation tasks.

SKR (Wang et al., 2023a) trains a classifier to estimate whether the LLM already has sufficient internal knowledge to answer a query. The classifier is trained using model outputs with and without retrieval. Based on this prediction, the retriever is selectively invoked. This approach balances internal and external knowledge usage. Its effectiveness depends on the reliability of the classifier, which may be affected by label bias.

SELF-RAG (Asai et al., 2024) integrates reflective capabilities into the LLM for retrieval-aware generation. The model learns to self-assess when retrieval is needed, how relevant the retrieved evidence is, and whether the final output is reliable. These signals are encoded using dedicated reflection tokens during training. The approach allows finer-grained control over the retrieval process. It enhances generation quality without relying on external classifiers.

**Adaptive RAG**, Adaptive-RAG and RQRAG adjust the retrieval depth or formulation based on question complexity or model uncertainty.

Adaptive-RAG (Jeong et al., 2024) dynamically selects among multiple retrieval strategies based on query complexity. A lightweight classifier predicts whether a query is simple, moderate, or complex. Depending on the classification, the model chooses to skip retrieval, perform single-step retrieval, or use multi-step retrieval. Training labels are automatically derived from model predictions and corpus-level patterns. This reduces the need for manual annotation while improving retrieval efficiency.

RQ-RAG (Chan et al., 2024) enhances the retrieval process through query optimization techniques. It applies rewriting, decomposition, and disambiguation to refine the original query. These improved queries lead to higher-quality document retrieval. The approach increases robustness to query noise and corpus mismatch. It demonstrates the critical role of query formulation in retrieval-augmented generation.

Iterative RAG and Iter-RetGen perform multi-round retrieval guided by intermediate generation hypotheses to progressively refine the answer context. This taxonomy enables a comprehensive evaluation across different retrieval paradigms.

ITER-RETGEN (Shao et al.) introduces an iterative coordination between retrieval and generation. The model first generates an initial answer draft, which is used to trigger a second retrieval. The retrieved results then guide regeneration of the answer. This generate–retrieve–regenerate cycle is repeated to refine the response. The method reduces the number of retrieval calls while maintaining strong performance.

## C  CASE STUDY

To gain a deeper understanding of how DynaRAG enables dynamic retrieval-augmented generation in RAG scenarios, we conducted a case study combined with manual analysis. Table 9 presents questions that can be answered using the language model's internal knowledge, while Table 10 lists those that require external retrieval support.

---

### Case 1: Direct Answer

---

**Question:** What timezone is Sweden?
**Answer:** ["Central European Time"]
**KRE**
**Instruction:** Provide relevant knowledge to answer this question.
**The problem is:** What timezone is Sweden?
**Output:** Sweden is in the Central European Time (CET) zone during standard time (UTC+1) and observes Central European Summer Time (CEST) during daylight saving time (UTC+2).
**s (Similarity Computation Score) >t (Threshold)**
**Instruction:** Answer the question using one's own internal knowledge.
The problem is:What timezone is Sweden?
**Output:** Sweden is in the Central European Time (CET) zone.

---

### Case 2: Retrieval-Augmented Generation

---

**Question:** Where does the director of the film *Ghost in the Machine* (Film) work?
**Answer:** ["McGill University College of British Columbia", "Ubc", "University of British Columbia", "UBC", "The University of BritishColumbia"]
**KRE**
**Instruction:** Provide relevant knowledge to answer this question. **The problem is:** Where does the director of the film *Ghost in the Machine* (Film) work?
**Output:** The director of *Ghost in the Shell* (2017), Rupert Sanders, is a Hollywood filmmaker based in Los Angeles, where the film was produced by DreamWorks and Paramount. For the 1993 film *Ghost in the Machine*, director Rachel Talalay worked primarily in the U.S. and Canadian film industry.
**s (Similarity Computation Score) <t (Threshold)**
**KDE**
**Instruction:** Generate different similar questions based on the given question.
**The problem is:** Where does the director of the film *Ghost in the Machine* (Film) work?
**Output:** 1. Which institution is Rachel Talalay currently affiliated with in her academic or professional work? 2. What university does the director of *Ghost in the Machine* teach at? 3. Is Rachel Talalay a professor at any Canadian or American university?
**Retrieve**
*Document1:* Rachel Talalay is currently a professor in the Department of Theatre and Film at the University of British Columbia (UBC), where she teaches film production and direction.
*Document2:* Talalay has worked extensively in both Hollywood and Canadian television and film. Since the early 2010s, she has held a faculty position at UBC in Vancouver. *Document3:* UBC's Faculty of Arts lists Rachel Talalay as a full professor with a focus on genre cinema, especially science fiction and horror.
**RAG**
**Output:** The director of *Ghost in the Machine* (1993), Rachel Talalay, currently works at the University of British Columbia (UBC), where she serves as a professor in the Department of Theatre and Film.

---

