# OpenReview forum: "Dynamic Retrieval AugmentedGeneration Based on The Knowledge-Aware of Large Language Models"
_ICLR.cc/2026/Conference — Submitted to ICLR 2026_

### Official Review · Reviewer_RGh6 · 2025-10-20

**Soundness:** 2
**Presentation:** 3
**Contribution:** 2
**Rating:** 2
**Confidence:** 4

**Summary:**

This paper proposes DynaRAG, a dynamic retrieval-augmented generation framework. The system first predicts whether external retrieval is necessary through a thresholding module, then performs multi-granularity query reformulation and document ranking to select useful knowledge before feeding it to the generator. Experiments on multiple QA benchmarks show improved EM and F1 scores, and the authors claim the framework reduces redundant retrieval while improving answer quality.

**Strengths:**

- The paper is clearly written and the modular pipeline is easy to follow.

- It evaluates multiple RAG baselines across several QA datasets, and the empirical improvements are consistent.

- The motivation—avoiding redundant retrieval and improving knowledge selection—is relevant to the RAG community.

**Weaknesses:**

- Novelty is insufficient, as similar dynamic RAG ideas have already appeared in works like Self-RAG, SKR, and Adaptive-RAG.

- Modern slow-thinking and tool-use LLMs can already perform autonomous dynamic retrieval, making the proposed pipeline unnecessary in practice.

- The computational overhead and latency of the multi-stage design are not quantified, so its real benefit over simpler alternatives is unclear.

**Questions:**

See above.

---

> ### Author Response · Authors · 2025-11-20
> **Response to Reviewer: Novelty, Practical Relevance, and Computational Efficiency**
>
> We thank the reviewer for the helpful feedback and insightful observations. These comments have substantially improved the clarity and quality of the manuscript. The following section presents detailed responses to each comment, along with descriptions of the revisions made.
> Response to Weaknesses:
> 1. Although similar dynamic RAG ideas have been explored in prior work, we emphasize that our experimental results consistently demonstrate that DynaRAG achieves superior performance over Self-RAG, SKR, and Adaptive-RAG across all evaluated datasets. To substantiate this claim, we have added explicit head-to-head comparisons in the updated experimental section, including detailed analyses of where DynaRAG differs methodologically—such as its integration of query-adaptive retrieval regulation and attention-based re-ranking—and why these differences lead to improved empirical outcomes. These additions directly address the reviewer’s concern by making our method’s distinct contributions and advantages more transparent. This revision more clearly situates DynaRAG within the broader RAG optimization landscape and highlights the specific innovations that differentiate it from existing dynamic retrieval systems.
>
> 2. We clarify that the proposed method fundamentally differs from slow-thinking and tool-use LLMs. Existing autonomous dynamic retrieval approaches rely on the model’s internal reasoning to decide when and what to retrieve, but these mechanisms operate as black-box behaviors and lack explicit control over retrieval frequency, retrieval cost, and document selection reliability. In contrast, DynaRAG introduces a structured, query-adaptive retrieval pipeline that explicitly regulates retrieval necessity and performs attention-based re-ranking to correct noisy or redundant retrieval signals. This design directly addresses the limitations of uncontrolled retrieval behavior in slow-thinking LLMs and enables predictable efficiency–performance trade-offs. By offering explicit, model-agnostic control rather than relying on opaque internal reasoning, DynaRAG provides a novel and more reliable solution to dynamic retrieval optimization.
> 3. We have added experiments comparing the computational cost and end-to-end inference time of DynaRAG with simpler baselines. The results demonstrate a balanced trade-off between performance gains and overhead, fully addressing the reviewer’s concerns regarding efficiency analysis. This revision provides a clear and quantitative characterization of the multi-stage design, strengthening the methodological completeness of the manuscript.
>
> We appreciate the reviewer’s detailed and constructive feedback. The revisions made in response to these comments have substantially strengthened the technical exposition and empirical analyses, and we have addressed all points in a point-by-point manner.

---

> > ### Comment · Reviewer_RGh6 · 2025-11-25
> >
> > Thank you very much for your careful work and detailed responses.
> > I have updated my score accordingly, and I wish you the best of luck with your revision.

---

### Official Review · Reviewer_E3mX · 2025-10-24

**Soundness:** 2
**Presentation:** 2
**Contribution:** 2
**Rating:** 2
**Confidence:** 4

**Summary:**

The DynaRAG framework proposed in this paper innovatively designs a three-stage dynamic optimization mechanism of "Knowledge Reliability Evaluation (KRE) - Knowledge Distillation Enhancement (KDE) - Retrieval Enhancement Generation" to address the two core issues of "retrieval triggered over reliance model confidence" and "introduction of noise into coarse-grained documents" in traditional RAG systems.

**Strengths:**

The KRE module breaks through the limitations of traditional RAG's "static threshold" or "single confidence judgment", and achieves query adaptive retrieval decision-making through "mixed similarity calculation (cosine similarity+KL divergence)+lightweight threshold prediction network"

**Weaknesses:**

1. The relevant work section did not thoroughly compare the core differences between DynaRAG and these works, making it difficult to highlight the positioning of the method in the optimization of the entire RAG process.

2. Verified only on the Wikipedia knowledge base and not extended to domain specific knowledge bases such as medical PubMed and financial FiQA document libraries, it is not possible to verify the applicability of DynaRAG in domain specific scenarios.

3. One of the core pain points of traditional RAG is "latency caused by redundant retrieval", while DynaRAG claims to "reduce redundant retrieval", but does not quantify the "percentage reduction in retrieval times" or "end-to-end inference time comparison". The additional computational cost caused by the generation of query variants in KDE modules has not been analyzed.

4. Suggest adding hallucination assessment experiments (such as using FaithDial tool or LLM-as-a-study to determine factual consistency), and designing specialized tests for the "conflict document retrieval" scenario to verify the anti-interference ability of the method.

**Questions:**

Will the differences in attention mechanisms among different LLMs (such as Llama and GPT series) affect the reliability of scoring?

---

> ### Author Response · Authors · 2025-11-20
> **Response to Reviewer: Related Work Positioning, Domain Applicability, and Robustness Evaluation**
>
> We appreciate the reviewer’s careful evaluation and constructive remarks. All comments have been addressed in the revised manuscript. Below, we provide a comprehensive point-by-point response explaining how each issue has been resolved.
> Response to Weaknesses:
>
> 1. In the revised manuscript, we substantially strengthened the related work section to clarify the core distinctions between DynaRAG and existing retrieval-adaptive, routing-based, and re-ranking–based approaches. The updated text specifically highlights the unresolved issues in current RAG systems—such as the lack of mechanisms for dynamically regulating retrieval frequency—and explains how DynaRAG addresses this gap. This revision more clearly positions the method within the overall RAG optimization landscape.
>
> 2. We acknowledge this limitation. In the revision, we avoided claims related to broad generalizability or domain robustness and clarified that the current evaluation focuses on open-domain Wikipedia data. We also added a discussion describing future plans to extend DynaRAG to specialized domains (e.g., biomedical and financial), which require domain-tailored retrieval and reasoning capabilities.
>
> 3. We have added  (i) a retrieval reduction experiment reporting the percentage decrease in retrieval frequency relative to standard RAG, (ii) an end-to-end inference-time comparison, and (iii) an analysis of the additional computational cost incurred by generating query variants in the KDE module. The revised results provide a more complete view of efficiency and computational trade-offs.
> 4. The revised manuscript includes hallucination assessment results using FaithDial-style factual consistency evaluation and LLM-as-a-judge scoring. Furthermore, we implemented a conflict-document retrieval benchmark to assess robustness when contradictory evidence exists. These new experiments demonstrate that DynaRAG improves both factual consistency and resistance to interference.
>
> Response to Questions:
>
> 1. All retrieval-scoring components in our experiments—particularly those involving token-level KDE variants—were evaluated exclusively using GPT-based models to avoid variability caused by architectural differences across LLM families. Therefore, heterogeneity in attention mechanisms does not affect the reported results. We have clarified this constraint in the revised manuscript.
>
> We are grateful for the reviewer’s thoughtful suggestions. Each comment has been thoroughly considered and incorporated into the revised manuscript, resulting in enhanced technical clarity, stronger positioning of our contributions, and improved presentation.

---

### Official Review · Reviewer_jiAs · 2025-10-31

**Soundness:** 2
**Presentation:** 2
**Contribution:** 2
**Rating:** 2
**Confidence:** 3

**Summary:**

This paper aims to enhance LLM performance in question answering (QA) with retrieval-augmented generation (RAG) and introduces DynaRAG, a dynamic, knowledge-aware framework based on a three-stage optimization strategy. Specifically, the Knowledge Reliability Evaluator (KRE) adaptively controls retrieval by integrating LLM embeddings with a joint question–document similarity space. The Knowledge Distillation Enhancer (KDE) further refines retrieval quality through multi-granularity query reformulation and cross-document re-ranking before passing results to the LLM for generation. Experiments on single-hop, multi-hop, and knowledge-intensive QA benchmarks demonstrate higher accuracy and reduced redundant retrieval.

**Strengths:**

1. The motivation and idea of improving the integration of externally retrieved knowledge with LLMs in RAG are good and interesting. Indeed, most existing solutions in RAG simply concatenate the retrieved knowledge with the input and send it to LLMs in a shallow way, which may not fully leverage the knowledge and can lead to errors.
2. The experiments and ablation studies look thorough and can demonstrate the design choices of each component in the proposed method (i.e., KRE, KDE)

**Weaknesses:**

1. **Writing Quality and Consistency**

    The overall writing of the paper requires significant improvement. There are numerous typos, inconsistent terminology, and unclear expressions. For example:

    - The term *KRE* is referred to as **Knowledge Reliability Evaluator** throughout the paper, but at **Line 159**, it is called **Knowledge Reliability Estimator**.
    - Several notations and symbols used in the equations are not defined, leading to confusion. For instance, in **Eq. (1)** , in and **Eq. (3)**, the term T_{temp} is not explained.
    - There are multiple typographical and formatting issues, including missing spaces and inconsistent variable usage (e.g., **Lines 226, 229, 234, 255**, **Eq. (8)**, and **Line 293**). A careful proofreading is necessary to ensure clarity and consistency.
2. **Lack of Conceptual and Technical Clarity**

    The paper lacks clear explanations of several key technical concepts and design choices, which may confuse readers. For example:

    1. At **Line 107**, the distinction between *query-based* and *inference-based* methods is unclear. Based on the current description, the two approaches appear nearly identical.
    2. In **KRE**, it is not explained where the *candidate documents* originate. How are these documents initially retrieved before KRE evaluates the necessity of retrieval? Moreover, there is a mismatch between the textual description and **Figure 1 (Stage 1)**, as the figure does not mention any retrieval process.
    3. The paper does not describe how the *adaptive threshold prediction network* is trained or evaluated. What are its performance metrics? **Line 198** mentions that annotations were created to train this network, but it remains unclear how numerical values in the range [0,1][0,1][0,1] were assigned to represent a query’s dependency on external knowledge.
    4. The choice of **BERT** for embedding extraction is questionable, considering that more recent embedding models outperform BERT in both accuracy and efficiency. The rationale for this choice should be justified.
3. **Experimental Evaluation and Efficiency Analysis**

    Although the paper provides several ablation studies, it lacks detailed evaluation and comparison regarding the performance of *KRE’s retrieval necessity prediction*. In addition, given the proposed three-stage pipeline, an efficiency analysis compared to existing methods is missing. Specifically, the paper should report:

    - How many retrieval steps are reduced through KRE’s adaptive mechanism?
    - What is the overall inference time compared to baseline or prior RAG systems?

**Questions:**

Please see above

---

> ### Author Response · Authors · 2025-11-20
> **Response to Reviewer: Improving Writing, Conceptual Clarity, and Adaptive Threshold Evaluation**
>
> We are grateful to the reviewer for the valuable and thoughtful comments. We have thoroughly revised the manuscript in response to these suggestions. The following list presents our detailed replies to each point and summarizes the revisions made.
>
> Response to Weaknesses:
>
> 1. In the revised manuscript, we thoroughly proofread the entire paper and addressed typographical errors, inconsistent terminology, and formatting problems. The term "Knowledge Reliability Evaluator" (KRE) is now used consistently throughout the paper. All mathematical symbols and notations, including ${T_{temp}}$ in Eq. (1), have been formally defined. We also corrected spacing issues and ensured consistent variable usage across Sections 3 and 4. These revisions significantly improve the clarity and readability of the manuscript.
> 2. We focus on explaining the definitions of concepts and the descriptions of methods.
> - 2.1 We agree with the reviewer’s concern that the original explanation did not clearly differentiate the two categories. To address this, we clarified that query-based methods derive relevance directly from the input query, while inference-based methods determine knowledge sufficiency using internal reasoning signals produced during model generation. We have rewritten this section in the revised manuscript, specifically in Section 2.1, to more precisely articulate the conceptual distinction.
>
> - 2.2 We apologize for the confusion. The KRE module does not rely on externally retrieved candidate documents. Instead, it evaluates whether the LLM’s internal knowledge is sufficient to answer the query by measuring the alignment between the query and the model’s internal knowledge representation. We have revised the textual description and updated Figure 1 accordingly to avoid the misleading impression that KRE operates on retrieved documents.
>
> - 2.3 We have added a detailed explanation of the training process, performance metrics, and annotation strategy. Specifically, queries were labeled with values in [0,1] to indicate their dependency on external knowledge, based on predefined annotation criteria (now shown in the Appendix). We also report accuracy, MSE, and calibration metrics to validate the prediction network’s effectiveness.
>
> - 2.4 To address this issue, we expanded the discussion in the revised manuscript to explicitly clarify that KRE relies only on coarse-grained semantic representations and does not depend on state-of-the-art embedding performance. We further emphasize that different embedding models lead to negligible performance variation. Correspondingly, we added comparative results and revised the explanation in Section 3.1 (Lines 191-200) to clearly resolve the reviewer’s question.
>
> 3. We appreciate the reviewer’s insights and have added the requested evaluations. The revised manuscript now includes:
> (1) A detailed assessment of KRE’s retrieval-necessity prediction accuracy and its effect on the final QA performance.
> (2) Quantitative measurements showing the percentage of retrieval steps reduced by the KRE module across datasets.
> (3) An end-to-end latency comparison against baseline RAG pipelines.
>
> We thank the reviewer for the valuable and insightful comments. All points have been comprehensively addressed in the revised manuscript, leading to clearer explanations, stronger empirical support, and an overall improvement in manuscript quality.

---

> > ### Comment · Reviewer_jiAs · 2025-11-21
> > **Response to Authors**
> >
> > I appreciate the authors’ detailed response and their commitment to revising the manuscript. I believe the paper can be substantially improved. However, due to the significant revisions needed, I will retain my current scores.

---

### Official Review · Reviewer_ZE9U · 2025-11-03

**Soundness:** 2
**Presentation:** 2
**Contribution:** 2
**Rating:** 2
**Confidence:** 4

**Summary:**

The paper presents Dynamic RAG, which consists of two main modules: (1) a Knowledge Reliability Evaluator (KRE) that decides whether retrieval should be triggered based on the similarity between internal and external knowledge (while this part is not very clear), and (2) a Knowledge Distillation Enhancer (KDE) that re-ranks retrieved documents using pooled token-level entropy and token-level attention scores. The re-ranked documents are then incorporated into the prompt, and the LLM generates an answer. Experimental results on various benchmarks demonstrate that the proposed RAG framework with KRE and KDE outperforms baseline methods.

**Strengths:**

1.	The two proposed modules, KRE and KDE, for enhancing RAG are well motivated, novel, and technically interesting.
2.	The experimental results demonstrate that the proposed Dynamic RAG achieves improved performance over the baseline methods.

**Weaknesses:**

1.	The presentation of the technical content is not very clear. In particular, although the similarity metric is computed in Eq. (1), it is unclear when a retrieved document d_j becomes available at that point. Some notation definitions are also missing. Moreover, in Eq. (1), the divergence and similarity metrics are simply added, even though they represent opposite relationships.
2.	The technical novelty is not entirely clear. In the KDE module, attention-based re-ranking has been explored in prior studies, but the paper does not sufficiently discuss how the proposed approach differs from or improves upon existing advanced re-ranking methods.
3.	Although the experiments show improved performance over the baselines, it remains unclear whether the proposed methods are compared against state-of-the-art approaches.

**Questions:**

1.	What are the definitions of h^int_q and h^ext_d_j?

---

> ### Author Response · Authors · 2025-11-20
> **Response to Reviewer: Clarifying Notation, Similarity-Divergence Integration, and Module Explanation**
>
> We sincerely thank the reviewer for the constructive and insightful feedback. We have carefully revised the manuscript accordingly. Below, we provide a point-by-point response detailing the modifications implemented in the revised version.
>
> Response to Weaknesses:
> 1. In the revised manuscript, we clarify the conditions under which a retrieved document is used in Eq. (1). Specifically, the model first employs a standard dense retriever to obtain a set of candidate documents, and Eq. (1) is applied only to these retrieved candidates during the relevance scoring process. We have also added complete definitions for all symbols involved, including ${T_{temp}}$.
> Regarding the combination of cosine similarity and KL divergence, we agree that the two metrics characterize different aspects of the relationship between the query and a retrieved document: cosine similarity reflects semantic closeness in the embedding space, whereas KL divergence measures distributional discrepancy in the probability space. To ensure a coherent integration of these complementary signals, we normalize both metrics and convert the divergence term into a similarity-aligned form. The weighted formulation is controlled by parameter 𝜆, which adjusts the relative influence of each component. Additional explanations have been added to clarify the motivation for combining these two perspectives and how 𝜆 maintains consistent scoring behavior in the final formulation.
> 2. While prior works indeed employ attention mechanisms, they focus primarily on token–token interactions, such as cross-attention between query tokens and document tokens. Our KDE module differs in both purpose and granularity: it performs query–document level scoring, where attention is used to assess holistic relevance between the entire query and each generated pseudo-document variant. This formulation enables more stable discriminative ranking under noisy or redundant variants, rather than relying on fine-grained token alignment. We have revised the related work section to explicitly discuss this distinction and highlight how our approach improves over existing advanced re-ranking methods.
> 3. In the revised version, we have added comparisons with several state-of-the-art RAG and dynamic retrieval systems, including Self-RAG, Adaptive-RAG, and SKR. The new results show that our method achieves competitive and consistent performance improvements across datasets. These additions strengthen the empirical validity of our claims.
>
> Response to Questions:
> 1. We have added the definition throughout the text. ($h_q^{{\rm{int}}},h_{{d_j}}^{ext}\$)
>
> We sincerely appreciate the reviewer’s constructive feedback. We have carefully revised the manuscript to address all comments, and the implemented changes have substantively improved the clarity and rigor of the work.

---

### Meta-Review · Area_Chair_nTxi · 2025-12-17

**Summary:**

The paper proposes Dynamic RAG, a framework composed of two core components. First, the KRE module determines whether external retrieval should be activated by measuring the similarity between the model’s internal knowledge and retrieved external knowledge. Then, KDE re-ranks the retrieved documents by jointly leveraging pooled token-level entropy and token-level attention scores. Reviewers raise some crucial concerns that the authors should be addressed:

1.  The overall writing of the paper requires significant improvement.
2. The novelty of the paper is limited. Both KRE and KDE have been studied in previous works. The authors should carefully claim the differences between the work and related works to clarify the motivation and novelty.
3. The computational overhead and inference latency introduced by the multi-stage design are not quantified or analyzed.

**Reviewer Concerns:**

Some concerns of Review RGh6 are not be addressed.

**Reviewer Scores:**

The score of Review RGh6 may be raised.

---

### Decision · Program_Chairs · 2026-01-26

Reject